# Electrospun Ceramic Nanofibers for Photocatalysis

**DOI:** 10.3390/nano11123221

**Published:** 2021-11-27

**Authors:** Yan Xing, Jing Cheng, Heping Li, Dandan Lin, Yuting Wang, Hui Wu, Wei Pan

**Affiliations:** 1School of Science, Nanjing University of Posts & Telecommunications (NUPT), Nanjing 210023, China; xingyan618@njupt.edu.cn; 2State Key Lab of New Ceramic and Fine Processing, School of Materials Science and Engineering, Tsinghua University, Beijing 100084, China; chengj88@mail.sysu.edu.cn (J.C.); liheping@hust.edu.cn (H.L.); lindd04@163.com (D.L.); yutingwang@ustb.edu.cn (Y.W.); huiwu@tsinghua.edu.cn (H.W.)

**Keywords:** electrospinning, ceramic fibers, photocatalysts

## Abstract

Ceramic fiber photocatalysts fabricated by electrospinning hold great potential in alleviating global environmental and energy issues. However, many challenges remain in improving their photocatalytic efficiencies, such as the limited carrier lifetime and solar energy utilization. To overcome these predicaments, various smart strategies have been invented and realized in ceramic fiber photocatalysts. This review firstly attempts to summarize the fundamental principles and bottlenecks of photocatalytic processes. Subsequently, the approaches of doping, surface plasmon resonance, and up-conversion fluorescent to enlarge the light absorption range realized by precursor composition design, electrospinning parameter control, and proper post heat-treatment process are systematically introduced. Furthermore, methods and achievements of prolonging the lifetime of photogenerated carriers in electrospun ceramic fiber photocatalysts by means of introducing heterostructure and defective composition are reviewed in this article. This review ends with a summary and some perspectives on the future directions of ceramic fiber photocatalysts.

## 1. Introduction

Electrospinning is a fascinating technique that allows the fabrication of continuous fibers with diameters down to a few nanometers [1,2,3]. Since Cooley issued the first electrospinning patent in 1902 [4], this technology did not receive much attention until the research boom of nanomaterials in the 1990s. Now, electrospinning is widely recognized as a simple, versatile, and scalable method for preparing both polymer and ceramic nanofibers [5,6,7,8,9]. Typically, a high-voltage source, solution reservoir, spinning nozzle, and grounded collector are the main components of electrospinning equipment [2]. By adjusting relevant parameters, such as the distribution of the electric field, viscosity and surface tension of precursor solution, the size and shape of the nozzle, and the collector structure, the microstructure and morphology of the fibers can be finely controlled [10,11,12,13]. Because the fiber products usually exhibit high porosity, good flexibility, large specific surface area, and a uniform diameter, they shows wide application prospects in the fields of filtering [14], biological tissue engineering [15], electronic devices [16], and photocatalysis [17].

Energy shortages and environmental pollution are urgent issues facing humanity, despite the fact that the solar energy reaching the Earth far exceeds the global energy consumption [18,19]. Semiconductor photocatalysis can utilize renewable solar energy to initiate chemical redox processes such as pollutant degradation and hydrogen generation, solving the environmental and energy issues in a green, low-cost, and sustainable way [20,21,22]. Since Fujishima and Honda discovered photo-assisted electrochemical water oxidation on TiO_2_ electrode in 1972 [23], photocatalysts based on semiconductor materials have been studied extensively [24,25,26]. Numerous research groups have evolved to synthesize oxide semiconductor catalysts with different scales, morphology, and structure, and characterized their photocatalytic properties [27,28,29]. Among them, 1D ceramic fibers prepared by electrospinning of precursor nanofibers followed with a proper heat treatment process have attracted much attention. Compared with conventional nanoparticle photocatalysts prepared through complex chemical synthesis, the electrospinning method is direct and simple, and the prepared ceramic fibers possess a uniform morphology, high aspect ratio, large surface-to-volume ratio, 3D open structure, and rich surface active sites [2]. All these characteristics contribute to the enhanced photocatalytic activity and recovery rate of photocatalysts. Moreover, various strategies to optimize the performance of photocatalysts, such as doping, up-conversion illumination, and heterostructure and composite design, can be easily realized during electrospinning by simply changing the precursor prescription, electrospinning parameters, or thermal treatment process. Therefore, taking advantage of flexibility, easy recycling, and low cost, electrospun ceramic fibers have great potential as high-performance photocatalysts.

The further application of photocatalytic technology is mainly restricted by two aspects: (1) the narrow light response range; and (2) the limited photogenerated carrier utilization. During the past decades, a variety of strategies have been employed to improve the photocatalytic efficiencies of electrospun ceramic fibers. To expand the range of light response and develop visible-light-response photocatalysts, doping [30], solid solutions [31], the surface plasma effect [32], and up-conversion luminescence [33] have been widely used. In terms of improving the carrier utilization, the separation of photogenerated carriers is usually promoted by designing the heterojunction structure [34], the surface defect [35,36], and the facet effect and polarization [37,38]. These methods all can be realized in electrospun ceramic nanofibers, and we have developed a serial of novel ceramic fiber photocatalysts, including Ag-ZnO nanofibers [17], ZnO-ZnS core-shell nanofibers [39], and carbon quantum dots-TiO_2_ nanofibers [40]. We believe that electrospun ceramics fibers have great potential for photocatalytic applications and a comprehensive review can promote further developments in such a challenging field. This review focuses on the main obstacles in photocatalysis, proposes targeted solutions, and summarizes recent progress on electrospun ceramic fiber photocatalysts.

## 2. The Fundamental of Photocatalysis

From the perspective of photochemistry, semiconductor photocatalysts can achieve the aims of degradation, water splitting, and hydrogen production by initiating a serial of redox reactions under irradiation [2,18]. However, these photo reactions strongly depend on the intrinsic physical properties of the semiconductor, including the band gap and band position [27,41,42]. As illustrated in Figure 1, only a semiconductor with a proper band gap (*E*_g_) can absorb photons whose energy is equal to or greater than their band gap energy. Then, the electrons (e^−^) at the valance band (VB) will be excited to the conduct band (CB), leaving an equal number of holes (h^+^) at the VB. The photogenerated e^−^ and h^+^ at CB and VB hold different chemical potentials according to the band position and will participate in the subsequent reduction and oxidation reaction, respectively [43]. For example, if the chemical potential of VB is lower than that of OH^−^/·OH reaction, the holes at VB can react with absorbed OH^−^ and produce hydroxyl radicals with high oxidation activity for pollutant decomposition [44]. Similarly, when the CB holds a higher chemical potential than that of H^+^/H_2_, hydrogen can be generated with the assistance of photogenerated electrons [45]. During the photocatalytic processes, two issues should be noted. First, the band gap determines the range of the solar spectrum utilized by semiconductors, and wide band gap semiconductors such as TiO_2_ (*E*_g_ = 3.2 eV) can only be excited by ultraviolet (UV) light, which occupies 4% of the solar spectrum. Second, the photogenerated e^-^ and h^+^ pairs recombine easily and rapidly, accompanied by heat or light emission. Therefore, developing visible-light-response semiconductor photocatalysts and improving the utilization of photogenerated carriers still pose great challenges.

Taking advantage of the high aspect ratio, good flexibility, proper surface area, and ease of recycling, 1D nanofibers are attractive for practical applications as photocatalysts [46,47]. For the production of high-quality 1D nanofibers, electrospinning has been widely considered as a simple, scalable, and cost-effective technology. Moreover, the two bottlenecks in photocatalysts, namely the visible light harvesting and carrier utilization, have been well resolved in numerous electrospun ceramic nanofibers, as listed in Table 1. Among these, various strategies are employed during the electrospinning process to adjust the solar energy absorption range, which is introduced in Section 3. Furthermore, different heterostructure designs and defective engineering have been realized in electrospun ceramic fibers to prolong the carrier lifetime, as discussed in Section 4.

## 3. Ceramic Nanofibers for Visible-Light Harvesting

### 3.1. Principles of Electrospinning Ceramic Fibers

The setup of an electrospinning device is not complicated; typically, a high-voltage resource, syringe, metallic needle, and a grounded collector are enough [46], as shown in Figure 2. The precursor solution used for electrospinning should be uniform, non-conductive, and have an appropriate viscosity. Deionized water or ethanol are common solvents for precursor solutions, polymer provides a fiber skeleton and adjusts the solution’s viscosity, and soluble metal salt determines the composition of the product ceramic fibers. Direct current (DC) and alternating current (AC) power are both suitable for electrospinning [73]. The spinneret is connected to a syringe containing precursor solution. By placing the syringe vertically, the precursor solution can flow through the spinneret under gravity. In addition, using a syringe pump is an effective means to control the flow rate of the precursor solution. When a high voltage is applied, the pendant drop of precursor polymer solution at the spinneret becomes highly electrified and is then distorted into a Taylor cone under the electrostatic interactions. Once the electrostatic forces overcome the surface tension of the precursor solution, a liquid jet is ejected from the needle. This electrified jet then undergoes a whipping and solvent evaporation process, leading to the reduction in the fiber’s diameter from hundreds of micrometers to tens of nanometers [3]. Eventually, a non-woven fabric can be easily collected at the collector. For ceramic fibers, a subsequent thermal treatment process is needed to remove the polymer skeleton and other additives in the precursor solutions and obtain a well-crystallized structure.

Although the setup for electrospinning is simple, its mechanism is intricate and involves several physical instability processes [1]. Before 1999, the formation of nanofibers by electrospinning was ascribed to the repulsion of the charged liquid jet [74]. Now, experimental observations, electro-hydrodynamic theories, and mathematical model calculations all demonstrate that the thinning of a jet during electrospinning is mainly caused by the bending instability associated with the electrified jet [3,75,76,77]. In addition, its stability is determined by the electrostatic interactions between the external electric field and the polarized polymer solution [78,79,80]. The fibers with fine diameters can therefore be generated stably after the stretching and acceleration of the filament in the instability region. All these studies provide a better understanding of the electrospinning process and guidance for experimental design. By regulating the precursor composition, precursor viscosity, spinneret structure, electric field strength, and heat-treatment process, the morphology and microstructure of the prepared ceramic fibers can be finely designed and controlled. It is clear that electrospinning is an ideal choice for generating 1D nanostructures. However, many technical limitations still need to be resolved. First, electrospinning cannot effectively and quickly produce functional nanofibers for industrial manufacturing. Second, the location of fiber deposition is not easy to accurately control. Third, the production of separated fibers, especially single continuous long fibers, is hard to realize by electrospinning. Moreover, the strength of ceramic fibers is relatively low, and their mechanical properties need to be further optimized. Moreover, more studies are required in order to achieve a better control over the morphology and size of electrospun ceramic fibers. Nonetheless, electrospun ceramic fiber has achieved tremendous progress in recent years and shown application prospects in many fields, and particularly in that of photo catalysis.

For example, titanium dioxide is a pioneer of electrospun ceramic fiber photocatalysts, and is also a popular, excellent, and commercialized photocatalyst [43,81,82]. Li et al. selected tetrabutyl titanate, polyvinylpyrrolidone (PVP, Mw = 1,300,000), H_2_O, and acetic acid as starting materials, and calcinated the precursor fibers at 400 °C for 2 h in air to obtain anatase TiO_2_ nanofibers [48]. They found that the diameter of the nanofibers, controlled by changing the tetrabutyl titanate content in the precursor solution, played an important role in the photocatalytic activity of TiO_2_ nanofibers. With a decrease in the diameter from 245 to 205 nm, the photocatalytic activity of the TiO_2_ nanofiber is improved. However, further reduction in the diameter below 205 nm results in a deterioration in the photocatalytic activity, as shown in Figure 3. The TiO_2_ nanofibers with a diameter around 200 nm show a better performance, which is comparable with that of the commercial TiO_2_ photocatalyst nanopowders (P-25, Degussa Co., Ltd., Shanghai, China). In addition to TiO_2_, ZnO is also a representative photocatalytic material with a direct band gap (*E*_g_ = 3.4 eV), strong oxidation ability, and a large excitation binding energy, and can also be electrospun into nanofibers. Lin et al. prepared ZnO nanofibers by electrospinning and demonstrated its good photocatalytic property under UV light irradiation [17]. However, TiO_2_ and ZnO ceramic fibers need to absorb UV light to initiate the photocatalytic process. The search for novel visible-light-response ceramic fiber photocatalysts has been the main research trend in recent years.

### 3.2. Visible Light Responsive Ceramic Nanofibers

Efficient utilization of solar energy is essential for photocatalysts. The solar energy reaching the surface of the Earth (1.3 × 10^5^ TW), in which visible light is abundant, exceeds the global human energy consumption (16 TW) by nearly four orders of magnitude [18,83]. However, most of the semiconductor photocatalysts can only absorb ultraviolet light, which occupies 4% of the solar spectrum due to their wide band gaps. Therefore, to make better use of solar energy, it is important to identify a suitable semiconductor with a narrow band gap that can respond in the ultraviolet to visible wavelength range.

In recent years, a large amount of research has been devoted to finding visible-light-response photocatalysts, such as BiVO_4_ [64], Bi_2_WO_6_ [84], AgNbO_3_ [85], AgVO_3_ [86], Ag_3_VO_4_ [87], Ag_3_PO_4_ [66], CdS [67], V_2_O_5_ [88], and InVO_4_ [89]. Due to the formation of a continuous VB by Bi 6*s*, Ta 5*d*, or V 3*d* orbitals, these photocatalysts have a narrow band gap and can absorb visible light for photocatalysis. Fortunately, these promising visible-light-responsive materials can also be synthesized conveniently by electrospinning. Cheng et al. used bismuth nitrate, ammonium vanadate, citric acid, and PVP to prepare the precursor solution, and fabricated tetragonal sheelite (s-t) and monoclinic sheelite (s-m) phase BiVO_4_ nanofibers for the first time by electrospinning [61]. The band gap of the prepared BiVO_4_ nanofibers was around 2.55–2.58 eV, which is suitable for visible light irradiation. The results of RhB degradation demonstrated that more than 70% RhB was degraded by both single s-t and s-m phase BiVO_4_ nanofibers after 40 min visible light irradiation. A mat comprising electrospun Bi_2_WO_6_ nanofibers with a band gap of 2.68 eV exhibited enhanced photocatalytic activity in the decomposition of CH_3_CHO for CO_2_ evaluation. Shang et al. proposed that the hybridization of the Bi 6*s* and O 2*p* largely dispersed the VB, which facilitated the transfer of the generated charge carriers, thereby enhancing the photocatalytic oxidation of organic pollutants [90]. Although some achievements have been made in the search for visible-light-responsive materials, if the light absorption range of visible-light-responsive materials can be further expanded, or the light absorption range of ultraviolet-responsive materials can be extended to the visible light range, the utilization of solar energy will be greatly increased.

### 3.3. Approaches to Expand the Light Absorption Range

The light-absorbing range, limited by the intrinsic band gap, determines the theoretical solar energy conversion efficiency of semiconductors. To date, significant efforts have been made to expand the light-absorbing range of semiconductor photocatalysts from UV light to visible light. Doping, surface plasmon resonance, and up-conversion fluorescence have been widely proven to be efficient approaches and can be realized in electrospun ceramic fiber photocatalysts when controlling the precursor composition, electrospinning parameters, and the thermal treatment process.

#### 3.3.1. Precursor Composition Design

The precursor solution for electrospinning is composed of the soluble salt of target components, the polymer skeleton, and solvents. The choice of polymer, solvent system, and concentration have a great influence on the crystal structure, facet, and morphology of the electrospun ceramic fibers, thereby changing their photocatalytic performance. For example, Li et al. found that the diameter of titanium oxide fibers increases linearly with the increase in the concentration of tetrabutyl titanate (Ti(oBu)_4_) in the precursor solution [48]. Because the catalytic performance is closely related to the fiber diameter, the concentration can be used to control the performance. Regarding the polymer skeleton, there are various subtle designs, such as carbon-based ceramic fibers [63,66,67], carbon quantum dot (CQD) composite fibers [40], metal modified polymer fibers [49], and polymer modified ceramic fibers [91], that have been realized in electrospun fiber photocatalysts. Polyaniline (PANI) modified TiO_2_/polyacrylonitrile (PAN) composite nanofibers were prepared using a mixture of dimethylformamde (DMF) and N-methyl-2-pyrrolidone (NMP) as solvents [91]. The authors found that the PANI modified TiO_2_/PAN nanofibers demonstrated a relatively strong absorption in the visible region, whereas the pure PAN, TiO_2_ and TiO_2_/PAN nanofibers has no significant absorbing peak. Thus, up to 90% degradation of MO was achieved by the PANI modified TiO_2_/PAN nanofibers in less than 60 min in comparison with the neat nanofibers (about 10%). Moreover, the choice of salt is also important. Instead of the normally used nitrate, Lv et al. utilized the organic salt (Vanadium(IV)oxy Acetylacetonate) to induce the electrospinning gradient effect [51]. Then, hollow fibrous BiVO_4_ could be fabricated by single-spinneret electrospinning without a template. Benefiting from the unique tubular structure, BiVO_4_ nanotubes possess a hollow interior, leading to strong light harvesting ability and better photocatalytic performance than solid BiVO_4_ nanofibers. The authors proposed that this gradient effect is not subject to the usage amount, ratio, or kind of as-employed salt, which can also be applied for the fabrication of other tubular ceramic photocatalysts.

Furthermore, the form of liquid precursor is convenient for cation doping and solid solution design, and helps to uniformly disperse doped elements in the prepared ceramic fibers. Doping, including anion and cation doping, can tune the position of both CB and VB, thereby expanding the light absorbing range [92,93,94,95]. Cation doping, mainly metal ion doping, will introduce an impurity level in the semiconductor. Under light irradiation, the electrons at both the VB and impurity level can be excited to CB [96]. Transition metals, such as V, Fe, Co, Ni, Cr, and Mn, can use the 3*d* orbital to build a t*_2g_* impurity level and let the absorption edge of semiconductor red shift [97,98,99,100,101]. For example, Zhang et al. prepared a Fe^3+^-doped TiO_2_/SnO_2_ necklace-structured nanofiber by electrospinning [57,65]. The doping element Fe^3+^ and hybrid component SnO_2_ were introduced into the precursor solution by adding Fe(NO_3_)_3_·9H_2_O and H_2_SnO_3_ nanoparticles. The XRD results shown in Figure 4a proved that Fe was successfully doped into the TiO_2_ lattice with no third phase other than anatase and rutile appearing. They found that TiO_2_/SnO_2_ nanofibers with 0.5 at.% Fe doping possess the best photocatalytic performance, which exceeded that of non-doped nanofibers by a factor of more than 3 (Figure 4b). The author proposed that the 3*d* electron excitation of Fe^3+^ to the TiO_2_ conduction band can increase the visible light absorption, and thus enhance the photocatalytic performance. However, the impurity level also provides a recombination center for the photogenerated holes and electrons, which is not conducive to photocatalytic performance.

Through precursor solution design, the second phase particles can be introduced into the ceramic nanofibers, which can initiate surface plasmon resonance (SPR) [102,103,104]. The free electrons of the metal particles will resonate under light irradiation, resulting in strong visible light absorption [105]. Through this effect, the response range of semiconductor materials can be extended to visible light. Au, Pd, and Ag/AgBr are widely used as surface plasmon co-catalysts. Duan et al. prepared a Au/TiO_2_ nanofiber by electrospinning and demonstrated its photocatalytic activity by degradation of RB under visible light and nature light irradiation [58]. Gao et al. designed and fabricated the Z-scheme Au/TiO_2_/WO_3_ nanofibers [106]. They proposed that the SPR effect of Au can promote charge separation and absorption of visible light.

Electrospun ceramic fibers can also use the up-conversion luminescence effect to enhance the solar energy utilization. Ceramic fiber photocatalysts, containing rare earth elements and carbon quantum dots (CQDs) with an up-conversion luminescence effect, can absorb visible or even near-infrared light and emit ultraviolet or visible light [107,108,109]. Through this quantum tailor effect, the light response range of semiconductor materials can be further expanded. By intelligently designing the preparation process, these functional components can be introduced into the electrospun ceramic fiber. For example, Cheng et al. designed a CQD-TiO_2_ fiber photocatalyst based on the up-conversion luminescent effect [40]. This nanocomposite fiber was assembled by a simple electrospinning and impregnation process. First, TiO_2_ nanofibers were prepared by electrospinning and a subsequent calcination process. Then, glucose was converted to CQDs under the assistance of alkali and ultrasound. After immersing TiO_2_ nanofibers into the CQD solution, ultrasonic treatment began and the time was controlled to obtain the different thicknesses of carbon shells in which CQDs were dispersed (Figure 5a,b). Finally, a series of CQD-TiO_2_ fibers with carbon shells of about 1, 2, and 5 nm were obtained and labeled as C1, C2, and C3, respectively. They found that CQDs can absorb the near-infrared light and convert it into visible or even ultraviolet light. Then, the light in the range of 350–450 nm can excite TiO_2_ to start the photocatalytic reaction, as shown in Figure 5c. Based on this, the CQD-TiO_2_ nanocomposite fibers exhibited a remarkable enhancement in photocatalytic activity compared to commercial TiO_2_ powders (Degussa P25) and pure CQDs. The C2 samples had the best performance with complete RhB degradation achieved in 20 min of visible light irradiation (Figure 5d). Moreover, the CQD-TiO_2_ nanofibers demonstrated good repeatability during four continuous cycles shown in Figure 5e. The author proposed that the photocatalytic activity of the CQD-TiO_2_ nanofibers was highly enhanced by utilizing the rapid carrier separation via the carbon shells, up-conversion property of CQDs, and the heterostructure of TiO_2_ and amorphous carbon, as illustrated in Figure 5f. When the catalysts are illuminated by visible light with photon energy higher than the band gap of TiO_2_ and CQDs, electrons in the valance band can be excited to the conduction band with the generation of the same number of holes. Furthermore, the up-conversion property of CQDs enables the composite nanofiber to utilize more solar energy to increase the photocatalytic activity. Then, the photogenerated electrons and holes were separated through the intimate contact interface of TiO_2_ and carbon, and generated hydroxyl radical species and superoxide radical anions for the degradation of organic compounds. Their achievements indicate that the up-conversion effect can help the wide band gap photocatalysts expand the light absorption range which is also easily realized by electrospinning.

#### 3.3.2. Electrospinning Parameters

After completing the blend of the precursor solution, the variety of resulting fiber architectures becomes even greater with spinning parameter design. The main parameters are spinneret structure, number of nozzles, applied voltage, and tip to collector distance. By controlling the voltage and tip to collector distance, the intensity and distribution of the electrostatic field can be changed. Then, various microstructure containing soft surface, rough surface and beaded structure can be obtained in electrospun ceramic fibers. Lin et al. found that the jet diameter was a strong function of the applied voltage and the nozzle-to-ground distance and the formation of beads structure appeared with the increase in applied voltage [10]. Coaxial electrospinning is used for the preparation of core-shell nanofibers and hollow nanotubes [110]. In a typical coaxial electrospinning process, two concentric stainless steel needles are used for the inner capillary and outer tube to maintain the simultaneous separate feeding of two fluids. For example, TiO_2_ nanotubes were fabricated by coaxial electrospinning using the PVP solution as the core fluids and TiO_2_ sol as the shell [111]. Chang et al. analyzed the photocatalytic performance of hollow TiO_2_ nanotubes and solid TiO_2_ nanofibers by MB degradation and confirmed the superior performance of hollow TiO_2_ nanotubes [111]. The morphology and microstructure of the electrospun ceramic nanofibers can influence the light absorption, surface area, and surface active sites, such as the interior light interaction in hollow fibers, which enhances the light harvesting. Therefore, manipulating the microstructure of ceramic fibers during the electrospinning process by parameter control is an effective way to improve the photocatalytic performance.

#### 3.3.3. Heat Treatment Process

Heat treatment of electrospun precursor fibers is a key step to obtaining a ceramic fiber with good crystallization, controllable crystal phase, adjustable defects, and proper band gap. Calcination temperature, heating program, and atmospheric conditions are the main parameters. For example, Li et al. prepared a series of TiO_2_ nanofibers with different phase ratios by controlling the calcination temperature [60]. Wang et al. calcinated the precursor fibers under NH_3_ to prevent the transfer of the doping element Mn^2+^ to Mn^4+^ and synthesized a visible-light-response Mn-ZnO ceramic fiber photocatalyst [53]. Atmosphere sintering can also generate anion doping and surface defects, which have a great influence on photocatalytic activity. To date, (oxy)sulphides and (oxy)nitrides photocatalysts, such as TaON [112], Ta_3_N_5_ [113], Cu_2_ZnSnS_4_ [114], AlON [115], ZnIn_2_S_4_ [116], CdZnS [117], and Bi_2_O_2_S [118], have attracted ongoing interest due to their narrow band gaps and suitable band position. In contrast to introducing an impurity level by metal doping, as mentioned previously, N 2*p* and S 3*p* orbitals can help to form a valence band at potentials more negative than those of O 2*p* orbitals, resulting in a narrow band gap. Electrospinning can also be used to fabricate oxynitride nanofibers by controlling the calcination atmospheres. Li et al. annealed the electrospun TiO_2_ nanofibers under NH_3_ atmosphere and obtained surface nitride N-TiO_2_ nanofibers (Figure 6a) [55]. Compared with pure TiO_2_ nanofibers that can only be irradiated by UV light, the oxynitride nanofibers demonstrated good photocatalytic performance for RhB decomposition under visible light, as shown in Figure 6b. The author proposed that N-doping helps to expand the light absorption range from UV to visible light, and that the nitride temperature plays a key role. It is seen in Figure 6c that the reaction rate constant *k* of RhB degradation first increased with the nitridation temperature, then, when the temperature further increased above 500 °C, *k* decreased. The author proposed that this decrease in performance was ascribed to the formation of nonphotocatalytically active TiN phase. Cheng et al. prepared zinc-doped gallium oxynitride nanofibers by calcinating the electrospun Zn-Ga-O nanofibers under ammonia atmosphere [56]. The XRD results shown in Figure 7 confirm the formation of Zn-doped gallium oxynitride phase. They found that the absorption edge and band gap were affected by the degree of nitridation (Figure 7c,d). Compared with pure GaN (3.4 eV) and ZnO (3.2 eV), the band gap of oxynitride nanofibers was narrowed to around 2.7 eV when annealed at 850 °C, and the light absorption range was expanded to the visible light region. As expected, the oxynitride nanofibers exhibited excellent photocatalytic performance.

It can be seen that the electrospun ceramic fibers possess excellent photocatalytic activity and are suitable for the water treatment process. Their performance usually exceeds the particle catalysts with the same composition and the commercial TiO_2_ under the same conditions. Moreover, the electrospun ceramic fibers can be recycled easily and demonstrate stable performance during continuous tests, avoiding the problems of the energy-consumption regeneration process, catalyst leakage, and secondary pollution. Although doping, surface plasmon resonance, the up-conversion luminescence effect, and morphology have been used to narrow the band gap of electrospun ceramic fiber photocatalysts and thus enlarge the light-harvesting range, enhancing visible light absorption is still of vital importance for the development of new photocatalysts. The rise of high-entropy (HE) materials has led to new research directions [119]. HE materials have demonstrated excellent performance in the fields of thermal protection and insulation, rechargeable batteries, electrocatalysis, etc. By forming solid solutions with multi-principal elements, the band structure and band gap of HE ceramic fibers can be tuned and may be expanded to the visible light range, enabling the fibers to become high-performance photocatalysts. Hu et al. designed a high-entropy metal oxide containing 10 metal elements and achieved significant results in the field of electrocatalysts for oxygen reduction reactions [120]. However, there have been no reports concerning the photocatalytic properties of HE ceramic fibers. In addition, other approaches to expand the light-absorbing range are also worth studying for electrospun ceramic fibers.

## 4. Ceramic Nanofibers for Efficient Carrier Separation

When the light shines on electrospun ceramic fiber photocatalysts, electrons (e^−^) at the valance band (VB) will be excited to the conduction band (CB), leaving behind an equal number of holes (h^+^) at the VB, as illustrated in Figure 1. Then, the photogenerated carriers move to the surface of ceramic nanofibers, and stimulate and participate in the subsequent redox reactions for water splitting, pollutant degradation, etc. These carriers are critical to photocatalytic efficiency. However, their lifetime is extremely short, accompanied by dynamic recombination and annihilation. Thus, extending carrier lifetime and improving its utilization is another great challenge. At present, heterojunction construction, morphology design, and defect engineering are the main methods to separate the carriers in electrospun ceramic nanofibers.

### 4.1. Metal–Ceramic Heterojunction

The introduction of well-dispersed metal particles into the electrospun ceramic fibers forms a metal–ceramic heterojunction. The rectifying characteristics of a metal–semiconductor system can provide an efficient and feasible pathway for carrier separation [121]. For example, if the metal with a higher work function contacts a semiconductor (φ_m_ > φ_s_), electrons will pass from the semiconductor into the metal to equalize the Fermi energy levels and bend the energy band upward [122], as illustrated in Figure 8a. This process will form a positively charged space layer called the Schottky barrier which can serve as an efficient electron trap and prevent the recombination of electron-hole pairs. Conversely, the electrons will flow from the metal to the CB of the semiconductor, leaving behind a negatively charged space layer in which the band is bent downward (Figure 8b). Under light irradiation, the photogenerated electrons of the semiconductor can transfer to metal through the space charge layer; thus, the carrier lifetime can be prolonged. Based on this theory, various metal–semiconductor systems have been designed and realized by electrospinning, such as Ag-ZnO [17], Ag-BiVO_4_ [64], Ag-BaTiO_3_ [123], Au-TiO_2_ [124], and Pt/Au-BiVO_4_ [125].

Lin et al. added silver nitrate into the precursor solution and fabricated Ag-ZnO nanofibers by electrospinning [17]. During the electrospinning process, the precursor solution needed to be shaded to avoid decomposition of the silver nitrate. Furthermore, a plastic capillary was used as the spinneret, and the electric field was applied by dipping a charged silver thread directly into the precursor solution. They demonstrated that the Ag partials dispersed homogeneously along the ZnO nanofibers and formed a metal–semiconductor heterojunction (Figure 9a–d). Under UV light irradiation, the 7.5 at.% Ag-ZnO nanofibers exhibited better performance than ZnO nanofibers and commercial P25 with no decrease during the cycle tests, as shown in Figure 9e–h. The introduction of metal Ag facilitated the rapid transfer of the photogenerated electrons from ZnO to Ag, extended the lifetime of electron-hole pairs, and enhanced the photocatalytic performance. However, due to the wide band gap of ZnO, heterostructure design can only enhance the performance of Ag-ZnO nanofibers in the UV light absorption range.

To achieve visible light photocatalysis, Cheng et al. selected BiVO_4_ with a narrow band gap to construct a novel heterojunction with Ag [64]. Ag-BiVO_4_ nanofibers with different Ag contents were fabricated by electrospinning. Then, their photocatalytic activity was determined by dye degradation under visible light. As shown in Figure 10a, the RhB degradation performance was improved compared with pure BiVO_4_ nanofibers by introducing Ag particles. The photocatalytic activity of Ag-BiVO_4_ nanofibers increased with the increase in Ag content, and the composite fibers containing 10 at.% Ag showed the best performance, with almost 100% RhB degradation achieved in 20 min. By band structure analysis, they proposed that the photogenerated electrons migrated from CB of BiVO_4_ to Ag, impeding the recombination of photogenerated carriers (Figure 10b). Therefore, it is believed that the construction of a metal–ceramic junction can prolong the carrier lifetime and improve the photocatalytic properties of ceramic fibers.

### 4.2. Semiconductor–Semiconductor Heterojunction

In addition to the metal–semiconductor heterojunction, the heterojunction composed of two kinds of semiconductor can also be constructed easily in the electrospun ceramic fibers by controlling the precursor solution and thermal treatment process. According to the different band structure, there are two types of semiconductor–semiconductor (S-S) heterojunction, as illustrated in Figure 11. In the T-1 junction, the CB of semiconductor A is lower than that of B, whereas its VB is higher than that of B. After light irradiation, the electron and hole will transfer to A and stimulate a series of redox reactions. For example, Li et al. developed a special T-1 junction composed of rutile TiO_2_ and anatase TiO_2_ [60]. The anatase–rutile heterostructured TiO_2_ ceramic nanofibers were prepared by electrospinning. The fraction of rutile phase was controlled by the annealing temperature. The RhB decomposition performance of the obtained nanofibers was strongly dependent on the calcination temperature. The nanofibers annealed at 600 °C with 3 wt.% rutile phases exhibited the optimal performance, as shown in Figure 12a. The authors proposed that the transfer of electrons from the CB of anatase TiO_2_ to a lower energy CB of rutile TiO_2_ can increase the lifetime of carriers and thus enhance the photocatalytic efficiency (Figure 12b).

Various ceramic fiber photocatalysts with a type 2 heterojunction have been developed in recent years, including ZnO-ZnS core-shell nanofibers [39], tetragonal-monoclinic BiVO_4_ nanofiber [61], MnO-ZnO nanofibers [53], TiO_2_-SnO_2_ nanofibers [65], Ag_3_PO_4_-TiO_2_-CNFs [66], and Carbon–CdS/TiO_2_ nanofibers [67]. In a type 2 (T-2) heterojunction, the CB and VB of B are both higher than that of A, which can further derive two carrier transfer modes, as illustrated in Figure 11b,c. When the work function of component B is larger than that of A, a traditional T-2 heterojunction is formed and the electrons will flow from A to B to equalize the Femi level. This charge transfer process will build an inner electric field at the interface, as shown in Figure 11b, and provide an opposite transfer route for photogenerated electrons and holes. For example, Lin et al. synthesized the ZnO-ZnS core-shell nanofibers by electrospinning followed by a sulfurization process [39]. The thickness of the ZnS shell was controlled by the sulfurization reaction time (Figure 13a,b). They found that the 12 h sulfurization ZnO-ZnS nanofibers exhibited the optimal photocatalytic performance with a RhB degradation rate of 0.024 min^−1^, which is 10 times higher than that of pure ZnO, as shown in Figure 13c. They proposed that the ZnO and ZnS formed the T-2 type heterojunction (Figure 13d). Under UV light irradiation, the photogenerated electron of ZnS moved to the CB of ZnO, and the holes at the VB of ZnO transferred to the VB of ZnS, as illustrated in Figure 13d. The reverse transmission of electrons and holes can increase the yield and lifetime of carries.

For visible-light-response ceramic photocatalysts, Cheng et al. successfully prepared a novel BiVO_4_ nanofiber with tetragonal-monoclinic phase junction by electrospinning [61]. The phase ratio was controlled by the calcination temperature. The BiVO_4_ nanofibers calcinated at 450 °C exhibited the best photocatalytic activity under visible-light irradiation with an optimal rate constant and apparent quantum efficiency that was 10 times higher than that of the single tetragonal phase BiVO_4_ and 5 times higher than that of the monoclinic phase BiVO_4_ (Figure 14a–d). They revealed that the enhanced photocatalytic activity can be attributed to the T-2 type heterojunction formed in the electrospun tetragonal-monoclinic BiVO_4_ nanofibers, which prevented the recombination of electron-hole pairs and improved carrier utilization (Figure 14e). Similarly, the mechanism of the T-2 type heterojunction was also demonstrated in the MnO-ZnO nanofibers [53] and TiO_2_-SnO_2_ nanofibers [65] prepared by electrospinning. On the basis of the T-2 type heterojunction, the introduction of carbon as a support has been proven to provide a unique benefit in suppressing the charge recombination. Carbon-coated ceramics nanofibers, such as Ag_3_PO_4_-TiO_2_-CNFs [66] and carbon-CdS/TiO_2_ nanofibers [67], have been prepared by electrospinning and following the carbonization process, and present good photocatalytic performance.

Z-schema is the other kind of T-2 type heterojunction, where the work function of component B is less than that of A, as shown in Figure 11c. When they contact and form a heterojunction, the electrons flow from B to A and build an electric field towards A at the interface. Under light irradiation, the photogenerated electrons at the CB of A will flow to the VB of B through the space electric field at the interface, and maintain the highly redox-active electrons and holes at the CB of B and VB of A, respectively. The Z-type heterojunction provides a better solution for improving the charge separation efficiency and redox ability [112,113,114]. In recent years, direct and indirect Z-type photocatalysts have been developed, including LaFeO_3_/SnS_2_ [126], InVO_4_/AgVO_3_ [89], g-C_3_N_4_/MnO_2_ [127], and ZnO/CdS [128]. However, few studies have examined the construction of the Z-type heterojunction in electrospun ceramic fibers. Because the electrospun ceramic fibers possess better flexibility, are easily recycled and environmentally friendly, have rich surface active sites, and are low cost, it is expected that the electrospun Z-type ceramic nanofibers will show good photocatalytic performance, and is a promising future direction.

### 4.3. Defective Surfaces

Surface defect engineering has been found to be the critical factor to improve photocatalytic activity, and can trap the photogenerated electrons or holes and modify the phonon transport [129,130,131]. Taking advantage of doping and atmosphere heat treatment, different kinds of defects can be controllably introduced into electrospun ceramic fibers. For example, Cheng et al. introduced oxygen vacancy defects at the surface of electrospun (ZnO)_x_(GaN)_1−x_ nanofibers through atmosphere sintering and investigated their photocatalytic performance and mechanism [56]. The change in the concentration of RhB as a function of time is plotted in Figure 15a, which shows that the optimal nitridation temperature is 850 °C. After 100 min visible light irradiation, the RhB is almost completely degraded by the (ZnO)_x_(GaN)_1−x_ nanofibers and becomes colorless (Figure 15b). By dynamic calculation, the nanofibers calcined at 850 °C exhibit the highest rate constant *k* of 0.058 min^−1^ and apparent quantum efficiency (AQE) of 29.4%, which is about 40 times greater than that of the nanowires calcined at 600 °C (Figure 15c). Moreover, the electrospun (ZnO)_x_(GaN)_1−x_ nanofibers show very good stability in the cyclic test (Figure 15d). The authors confirmed the formation of oxygen and gallium vacancy, and proposed that the surface oxygen vacancy can trap photogenerated electrons and delay the recombination of the photogenerated electron-hole pairs, therefore enhancing the performance of the electrospun (ZnO)_x_(GaN)_1−x_ nanofibers, as shown in Figure 15e.

Wang et al. prepared Mn^2+^-doped and N-decorated ZnO nanofibers (NFs) enriched with oxygen vacancy defects by electrospinning [54]. To introduce a high density of oxygen vacancies and nitrogen, the precursor fibers were annealed under an atmosphere mixture of 50% air and 50% nitrogen at the beginning, and full nitrogen after heating to the temperature of 550 °C. The as-prepared fiber catalyst exhibited excellent visible-light photocatalytic activity and an apparent quantum efficiency up to 12.77%, which is 50 times higher than that of pure ZnO, as shown in Figure 16a,b. Through comprehensive structural characterization and calculation, they proposed that the oxygen vacancy formed mid-gap states, which serve as intermediate steps for the photoexcitation process. Moreover, these vacancy defects can capture electrons from the CB and form a neutral state (V_O_*), helping the holes release from the catalysts, and consequently delaying the recombination process (Figure 16c). Thus, the photocatalytic performance of the Mn^2+^-doped and N-decorated ZnO nanofibers was enhanced dramatically. Therefore, the introduction of defects can not only tune the band gap, but also prevent the recombination of photogenerated electrons and holes, providing an ideal solution to enhance the activity of ceramic fiber photocatalysts. However, the location, content, and state of defects are not easily controlled and characterized. These issues not only exist in photocatalytic materials prepared by electrospinning, and deserve further in-depth investigation.

## 5. Conclusions

Electrospun ceramic nanofibers have been extensively studied as photocatalysts for pollutant degradation and water splitting during the past decade. In this review, we summarized the typical ceramic nanofiber catalysts and recent progress for improving their performance, including: (1) the fundamental principles and bottlenecks of the photocatalysis process; (2) UV and visible-light-response ceramic fiber photocatalysts prepared by electrospinning; (3) the strategies used to expand the light absorption range of ceramic fiber photocatalysts; and (4) various types of heterostructure used for efficient carrier separation. At present, electrospun ceramic nanofibers with adjustable band structure have great application potential in the field of photocatalysis, and the electrospinning technique offers a scalable, easy, and low-cost route to produce ceramic fiber photocatalysts. Due to the efforts of many research groups, electrospinning has been developed as a technique capable of producing a variety of ceramics and composite fibers with fine structure and controllable morphology.

In the future, the photocatalytic efficiency and quantum yield of ceramic fiber photocatalysts should be enhanced for practical solar engineering. It is expected that novel strategies for improved light-harvesting and carrier separation will be realized in electrospun ceramic fibers. Moreover, the development of directly electrospun ceramic fabric or a mat with high solar energy conversion efficiency will become important, especially for the application to water splitting and hydrogen production. Regarding electrospinning technology, mass production of functional fibers with designed and controllable morphology will be one of the leading development directions. Then, combining the electrospinning techniques with other well-developed techniques for materials processing can increase the diversity of material and structure types, and open provide access to new properties and applications. As a result of the rapid innovation of science and technology, electrospinning will surely become an effective method for making 1D nanomaterials with a fine structure and ideal performance that is not limited to the field of photocatalysis.

## Figures and Tables

**Figure 1 nanomaterials-11-03221-f001:**
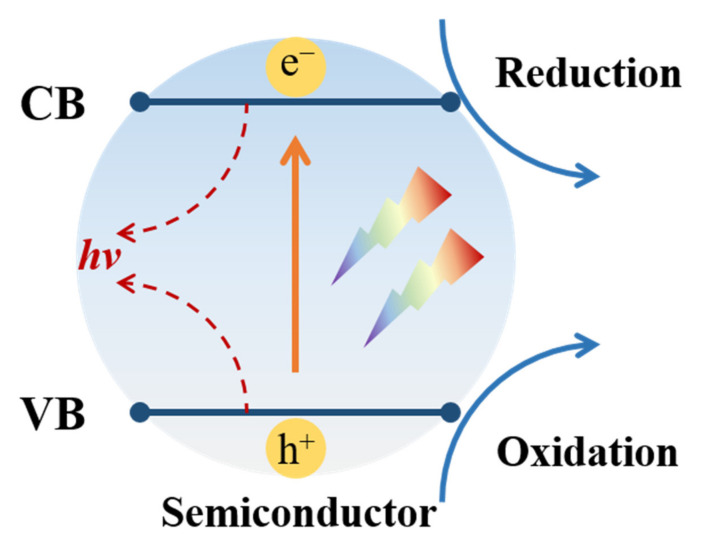
The diagram of semiconductor photocatalysis process.

**Figure 2 nanomaterials-11-03221-f002:**
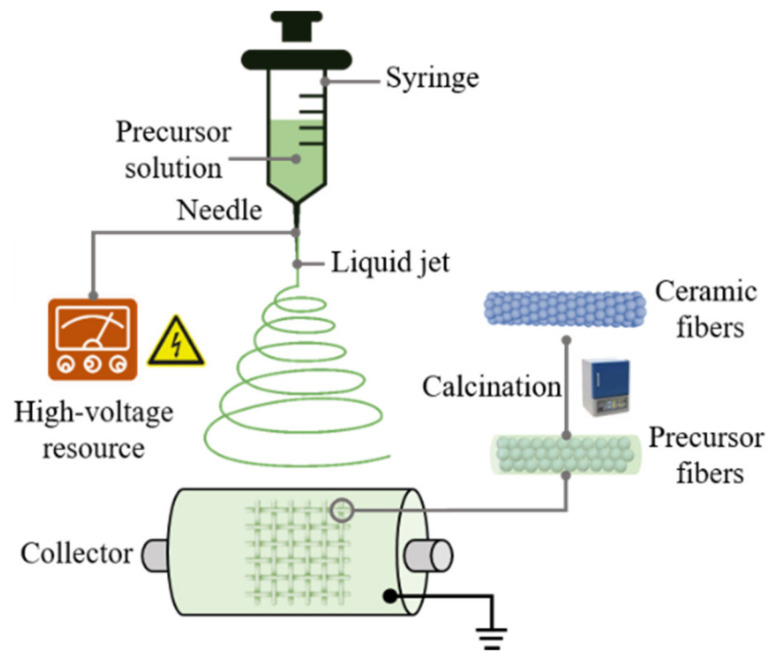
Schematic for the preparation of ceramic fibers by electrospinning.

**Figure 3 nanomaterials-11-03221-f003:**
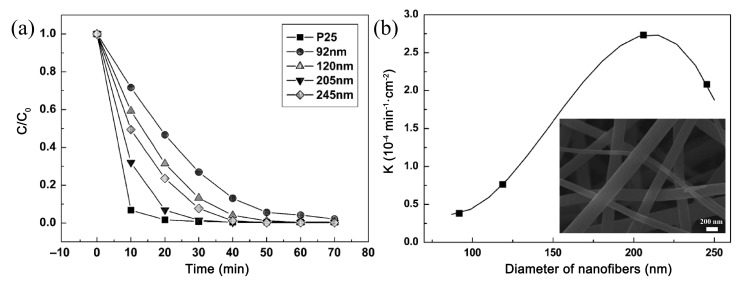
(**a**) Degradation of rhodamine B dye by TiO_2_ nanofibers with different diameters under UV light; (**b**) the relationship between the reaction rate constant and the diameter of TiO_2_ nanofibers (redrawn with the permission of Ref. [48]. Copyright 2010 John Wiley and Sons).

**Figure 4 nanomaterials-11-03221-f004:**
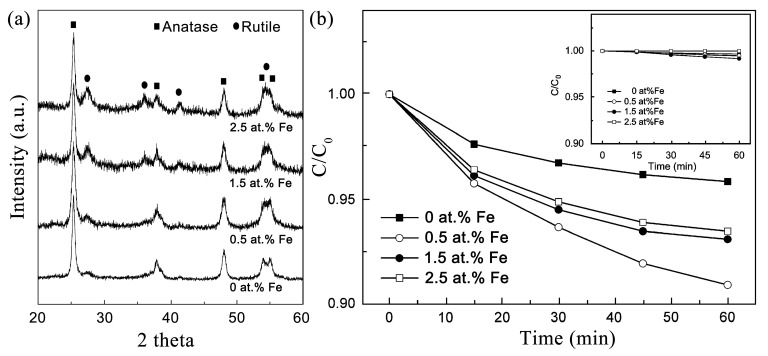
XRD results of Fe^3+^-doped TiO_2_/SnO_2_ nanofibers with different Fe content (**a**), and the degradation of RhB dye under visible light irradiation (**b**). Inset shows the adsorption of RhB on hybrid nanofibers. (Redrawn with the permission of Ref. [57]. Copyright 2010 John Wiley and Sons).

**Figure 5 nanomaterials-11-03221-f005:**
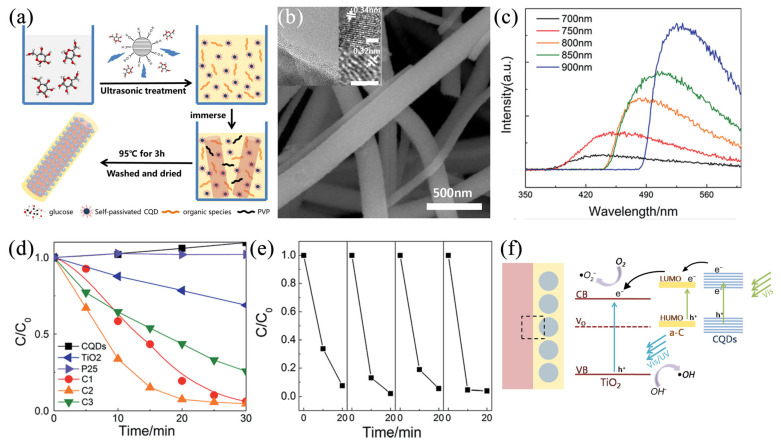
(**a**) Schematic illustration for the synthesis process; (**b**) the morphology of the CQD-TiO_2_ nanofibers; (**c**) the up-conversion properties of the CQDs; (**d**) photodegradation of RhB by the CQD-TiO_2_ nanofibers under visible light irradiation; (**e**) the repeatability tests; (**f**) schematic illustration of the photocatalytic mechanism of the heterostructure. (Redrawn with the permission of Ref. [40]. Copyright 2017 The Royal Society of Chemistry).

**Figure 6 nanomaterials-11-03221-f006:**
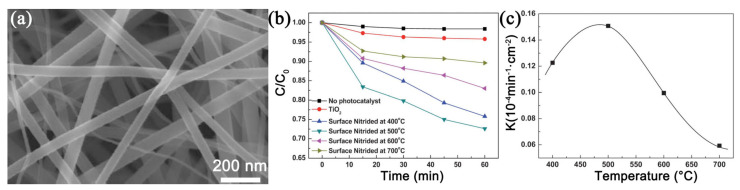
Morphology of N-TiO_2_ nanofibers nitrided under NH_3_ flow at 400 °C for 4 h (**a**); visible-light photocatalytic activity of N-TiO_2_ with different nitride temperatures (**b**); the relationship between the reaction rate constant and the nitride temperature (**c**). (Redrawn with the permission of Ref. [55]. Copyright 2012 The Royal Society of Chemistry).

**Figure 7 nanomaterials-11-03221-f007:**
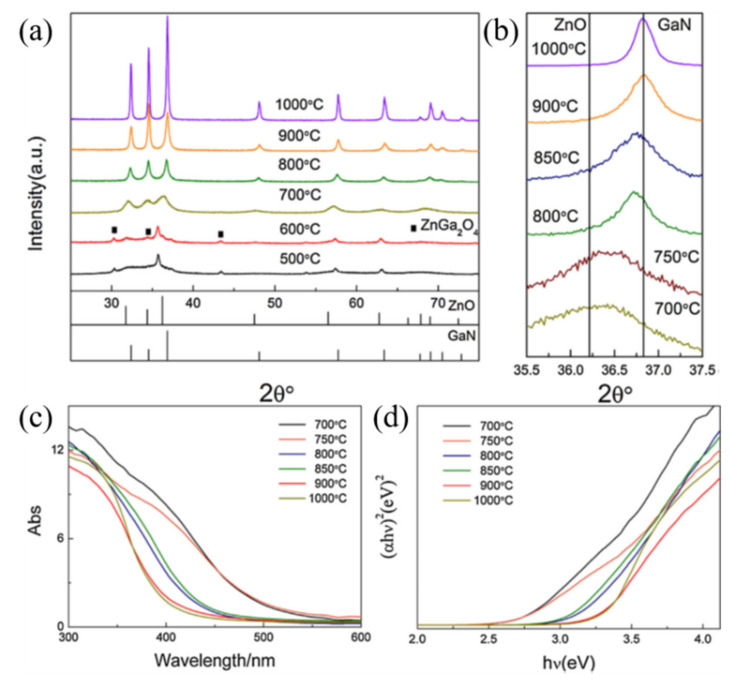
XRD patterns of samples calcined at different temperatures (**a**,**b**), UV–vis diffuse reflection spectra of the zinc-doped gallium oxynitride nanofibers calcined at different temperatures (**c**), and the corresponding plots of (*αhv*)^2^ vs. photon energy (*hv*) (**d**). (Redrawn with the permission of Ref. [56]. Copyright 2017 Elsevier).

**Figure 8 nanomaterials-11-03221-f008:**
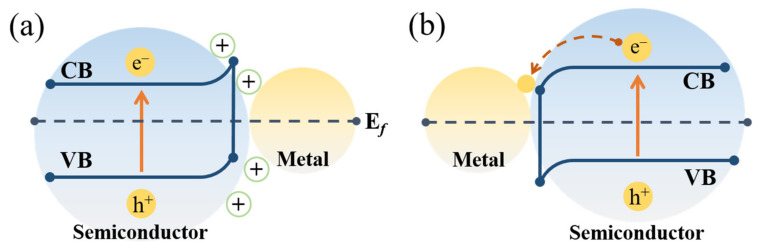
The schematic of the metal–semiconductor heterojunction: (**a**) φ_m_ > φ_s_ and (**b**) φ_m_ < φ_s_.

**Figure 9 nanomaterials-11-03221-f009:**
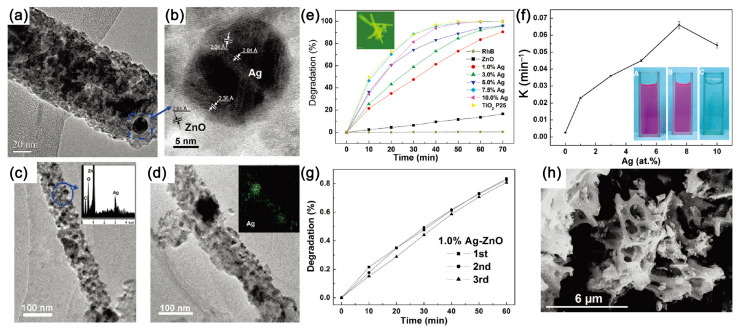
(**a**) The TEM image of 5.0 at.% Ag-ZnO nanofibers; (**b**) HRTEM image from the squared region of part a; (**c**) TEM image of 7.5 at.% Ag-ZnO nanofibers and EDX result on selected areas; (**d**) TEM image of 10 at.% Ag-ZnO nanofibers and EDX mapping of Ag element along the nanowire; (**e**) kinetics and (**f**) photodegradation rate constant of RhB by Ag-ZnO nanofibers under UV light; (**g**) the cyclic performance; (**h**) the morphology of Ag-ZnO nanofibers after photocatalytic measurement. (Redrawn with the permission of Ref. [17]. Copyright 2009 American Chemical Society).

**Figure 10 nanomaterials-11-03221-f010:**
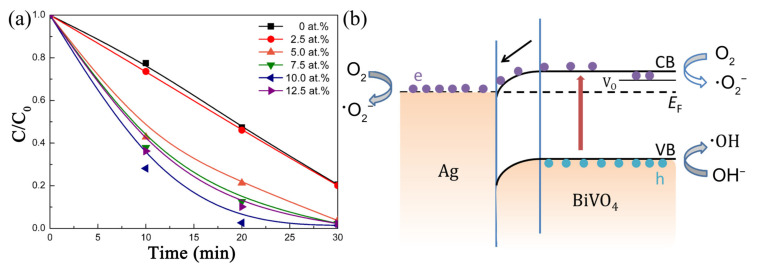
(**a**) The photocatalytic performance of Ag-BiVO_4_ nanofibers under visible light irradiation; (**b**) the proposed mechanism. (Redrawn with the permission of Ref. [64]. Copyright 2018 Science Press).

**Figure 11 nanomaterials-11-03221-f011:**
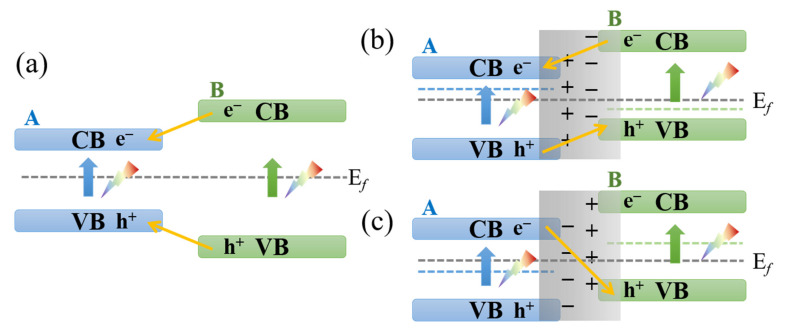
The illustration of semiconductor–semiconductor heterojunction type-1 (**a**) and type-2 (**b**,**c**), where A and B both represent semiconductors with different band structures.

**Figure 12 nanomaterials-11-03221-f012:**
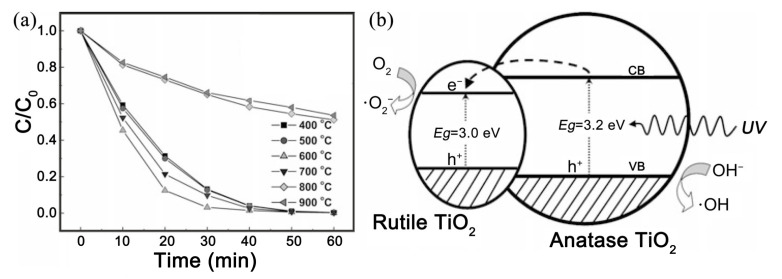
Photocatalytic performance (**a**) and proposed mechanism (**b**) of anatase–rutile heterostructured TiO_2_ ceramic nanofibers for dye degradation under UV light. (Redrawn with the permission of Ref. [60]. Copyright 2021 John Wiley and Sons).

**Figure 13 nanomaterials-11-03221-f013:**
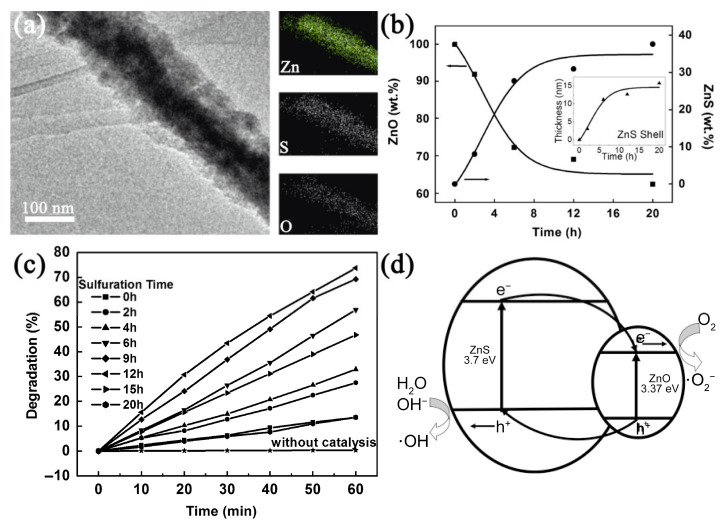
(**a**) The morphology and elementary distribution of ZnO-ZnS nanofibers; (**b**) dependence of phase contents on sulfuration times (inset shows the relationship between the thickness of the ZnS shell and the sulfurating time); (**c**) kinetics of the photodegradation of RhB by ZnO–ZnS nanofibers that were sulfurized for various times; (**d**) proposed schematic illustration of the band structure-related photocatalytic mechanism of the ZnO–ZnS heterostructure nanofibers. (Redrawn with the permission of Ref. [39]. Copyright 2010 John Wiley and Sons).

**Figure 14 nanomaterials-11-03221-f014:**
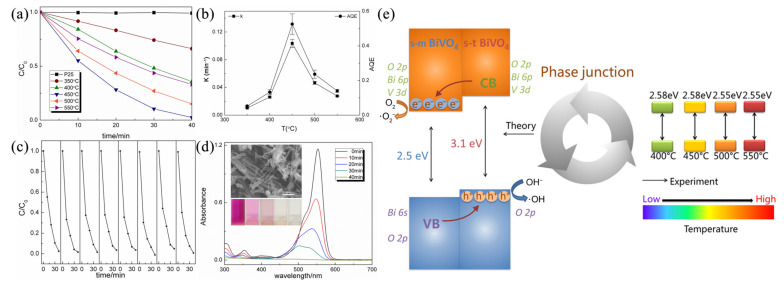
(**a**) Photodegradation of RhB by BiVO_4_ nanofibers calcined at different temperatures; (**b**) rate constants and apparent quantum efficiencies for BiVO_4_ nanofibers; (**c**) the repeatability tests of the specimen calcined at 450 °C; (**d**) the absorption spectrum of the RhB solution in the presence of BiVO_4_ nanofibers. The inset illustrates photos of the RhB solutions photodegraded with different time and the SEM image of the specimen reclaimed after photocatalytic measurement; (**e**) schematic illustrations of the band structure-related photocatalytic mechanism for the s-m/s-t BiVO_4_ phase junction. (Redrawn with the permission of Ref. [61]. Copyright 2015 The American Chemical Society).

**Figure 15 nanomaterials-11-03221-f015:**
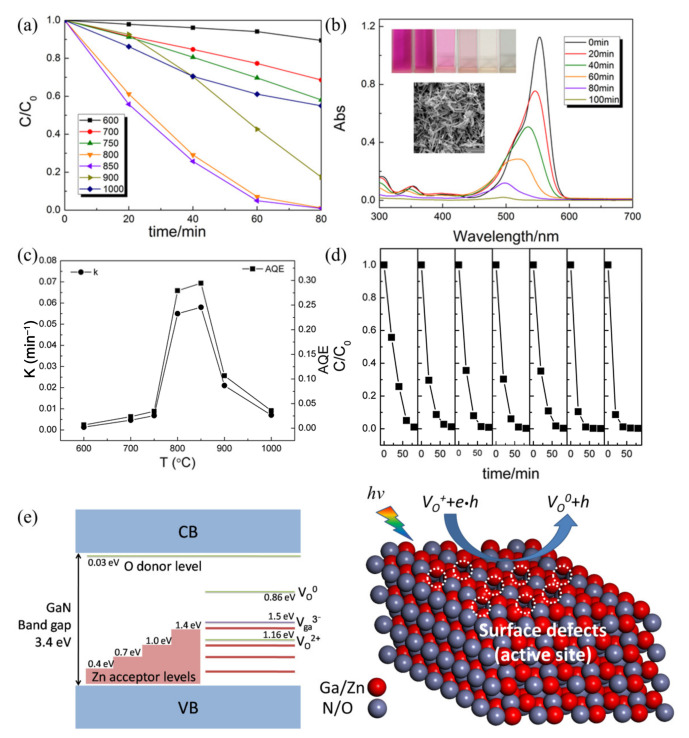
(**a**) Photodegradation of RhB by zinc-doped gallium oxynitride nanowires calcined at different temperatures; (**b**) the absorption spectrum of the RhB solution in the presence of the nanowires. The inset illustrates photos for the RhB solutions photodegraded for different times and the SEM image of the specimen reclaimed after photocatalytic measurement; (**c**) degradation rate constants and apparent quantum efficiencies for the nanowires; (**d**) the repeatability tests of the specimen calcined at 850 °C; (**e**) schematic illustrations of the surface defect-related photocatalytic mechanism for zinc-doped gallium oxynitride nanowires. (Redrawn with the permission of Ref. [56]. Copyright 2017 Elsevier ).

**Figure 16 nanomaterials-11-03221-f016:**
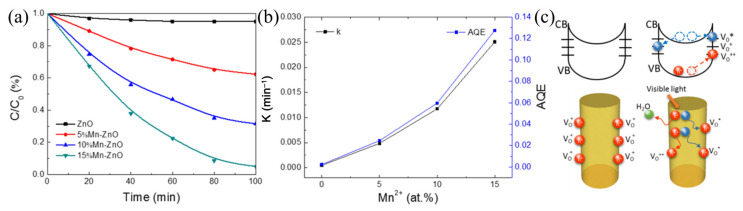
(**a**) Photodegradation of RhB by ZnO nanofibers doped with different Mn^2+^ concentrations. (**b**) Degradation rate constants and apparent quantum efficiencies. (**c**) Schematic of trapping and photocatalytic mechanism in a single NF under dark and visible light. (Redrawn with the permission of Ref. [54]. Copyright 2016 Wang, Y. et al.).

**Table 1 nanomaterials-11-03221-t001:** Summary of recent advances in ceramic fiber photocatalysts fabricated by electrospinning.

Photocatalyst	Strategy	Light Source	Experimental Conditions	Performance Efficiency/Time	Reference
TiO_2_	Diameter control	UV lamp	Catalyst: 0.1 wt.% RhB: 2.5 × 10^−5^ mol/L	~90%/20 min	[48]
Bimetal-PANNM	Surface area	UV lamp	Catalyst: 4 g/L Reactive blue 19: 25 mg/L	100%/80 min	[49]
CuWO_4_	Morphology	Visible light	Catalyst: 1 g/L MO: 5 mg/L	90%/180 min	[50]
BiVO_4_	Microstructure (Hollow fibers)	Visible light	Catalyst: 0.5 g/L Cr(VI): 10 mg/L	95.3%/80 min	[51]
ZnO	Defect Microstructure	UV light	Catalyst: 1 g/L MB: 1.5 × 10^−5^ mol/L	90%/85 min	[52]
Mn^2+^/ZnO	Doping	Visible light	Catalyst: 1 g/L RhB: 2.5 × 10^−5^ mol/L	~80%/260 min	[53]
N-Mn/ZnO	Doping Defect	Visible light	Catalyst: 1 g/L RhB: 2.5 × 10^−5^ mol/L	~95%/100 min	[54]
N-TiO_2_	Doping Defect	Visible light	Catalyst: 0.1 wt.% RhB: 2.5 × 10^−5^ mol/L	25%/60 min	[55]
(ZnO)_x_(GaN)_1−x_	Doping Defect	Visible light	Catalyst: 1 g/L RhB: 2.5 × 10^−5^ mol/L	~100%/80 min	[56]
Fe-TiO_2_/SnO_2_	Doping Heterostructure	Visible light	Catalyst: 0.05 wt.% RhB: 2.5 × 10^−5^ mol/L	10%/60 min	[57]
Au/TiO_2_	Surface plasmon resonance	Visible light	Catalyst: 0.4 g/L RhB: 10 mg/L	42%/120 min	[58]
Au/g-C_3_N_4_	Surface plasmon resonance	LED light	Catalyst: 2 cassettes MB: 5 ppm	~90%/50 min	[59]
TiO_2_	Phase junction	UV lamp	Catalyst: 1 g/L RhB: 2.5 × 10^−5^ mol/L	~100%/40 min	[60]
BiVO_4_	Phase junction	Visible light	Catalyst: 1 g/L RhB: 2.5 × 10^−5^ mol/L	100%/40 min	[61]
ZnO-ZnS	Heterostructure	UV lamp	Catalyst: 1 g/L RhB: 2.5 × 10^−5^ mol/L	73%/60 min	[39]
g-C_3_N_4_/TiO_2_	Heterostructure	Visible light	Catalyst: 1 g/L RhB: 5 mg/L	~75%/120 min	[62]
ZnO-TiO_2_-CNFs	Heterostructure	UV lamp	Catalyst: 0.8 g/L MB: 10 ppm	~95%/120 min	[63]
Ag-ZnO	Heterostructure	UV lamp	Catalyst: 1 g/L RhB: 2.5 × 10^−5^ mol/L	95%/40 min	[17]
Ag/BiVO_4_	Heterostructure	Visible light	Catalyst: 1 g/L RhB: 2.5 × 10^−5^ mol/L	~100%/20 min	[64]
TiO_2_/SnO_2_	Heterostructure	UV lamp	Catalyst: 0.1 wt.% RhB: 2.5 × 10^−5^ mol/L	90%/60 min	[65]
Ag_3_PO_4_-TiO_2_-CNFs	Heterostructure	Visible light	Catalyst: 0.5 g/L MB: 10 ppm	100%/10 min	[66]
Carbon-CdS/TiO_2_	Heterostructure	Visible light	Catalyst: 0.5 g/L MB: 10 ppm	100%/30 min	[67]
Bi_2_O_3_/g-C_3_N_4_	Heterostructure	LED light	Catalyst: 0.25 g/L TC: 20 mg/L	~60%/3 h	[68]
Co-CdSe@ECNFs	Heterostructure Doping	Visible light	Catalyst: 1 g/L MB: 10 mg/L	87%/90 min	[69]
ZnIn_2_S_4_/SnO_2_	Heterostructure Microstructure	Visible light	Catalyst: 0.6 g/L Cr(VI): 50 ppm	100%/80 min	[70]
TiO_2_@Ag@Cu_2_O	Heterostructure Surface plasmon resonance	Visible light	Catalyst: 0.5 g/L MB: 10 mg/L	99%/90 min	[71]
ZnFe_2_O_4_/Ag/AgBr	Heterostructure Surface plasmon resonance	LED light	Catalyst: 1 g/L RhB: 100 mg/L	86%/100 min	[72]
CQDs-TiO_2_	Up-conversion luminescence Heterostructure	Visible light	Catalyst: 1 g/L RhB: 2.5 × 10^−5^ mol/L	100%/20 min	[40]

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
