# Peer review of "Electrospun Ceramic Nanofibers for Photocatalysis"

_nanomaterials, 2021, doi:10.3390/nano11123221_

Round 1

Reviewer 1 Report

  1. Fig. 2 does not explain the fabrication protocol of ceramics nanofibers. Please, add the calcination process to it.
  2. Line no 147, more examples of visible light active photocatalysts should be given. For example, Ag3PO4, Ag3VO4, CdS,…..
  3. Carbon support is claimed to provide a unique benefit in suppressing the charge recombination of photocatalysts. Recently, carbon-coated ceramics nanofibers have been prepared by electrospinning/carbonization for photocatalysis. The authors should cover the following studies.

- https://doi.org/10.1016/j.seppur.2021.119400

- https://doi.org/10.1016/j.jcis.2014.07.039

  1. Besides the heat treatment, there are several parameters that can control the morphology of the ceramic nanofibers, thereby affecting the overall performance in photocatalysis. These parameters such as choice of polymer, concentration, conductivity, applied voltage, solvent system, tip to collector distance…………… should be explained.
  2. It would be better to provide a list of recent studies about electrospun ceramics nanofibers and their photocatalytic applications in a table.
  3. The limitations of electrospinning technology and challenges in Ceramics Nanofibers should be mentioned.
  4. The conclusion should cover future perspectives of ceramics nanofibers in photocatalytic applications.

Reviewer 2 Report

In this manuscript, Xing and coworkers report the photocatalytic applications of electrospun and heat-treated inorganic nanofibers. Especially, the authors described the efficient visible light absorption and facile charge separation strategies, in addition to the conventional introduction of photocatalysis and electrospinning process. Overally, this manuscript is carefully well-written. My opinions are as follows.

  1. Now, the electrospinning and the fiber fabrication is well-established and well-recognized among the scientists. However, I agree that the photocatalytic application aspect of electrospun inorganic fibers is a good review subject. In this regard, the present review manuscript will be good for the Nanomaterials.
  2. The scientific principles and strategies to improve the photocatalytic performance in this manuscript are conventional and those are common in inorganic photocatalytic materials. In my opinion, the metal doping, semiconductor heterojunction, and defect effects are very common in inorganic photocatalysts. Considering the intention of authors, are there special benefits of the electrospun inorganic photocatalysts in the aspect of principles or in the aspect of synthetic methods? If the authors can supply the special benefits of the electrospun inorganic photocatalysts, compared with the conventional inorganic photocatalysts, those will be very helpful for the readers.
  3. If the authors can add the summary table for the reported electrospun inorganic photocatalysts, features, strategies, and the comparative performance, those will be very helpful to follow the contents of this manuscript.

Author Response

Response to Reviewer 2 Comments

In this manuscript, Xing and coworkers report the photocatalytic applications of electrospun and heat-treated inorganic nanofibers. Especially, the authors described the efficient visible light absorption and facile charge separation strategies, in addition to the conventional introduction of photocatalysis and electrospinning process. Overally, this manuscript is carefully well-written. My opinions are as follows.

Point 1: Now, the electrospinning and the fiber fabrication is well-established and well-recognized among the scientists. However, I agree that the photocatalytic application aspect of electrospun inorganic fibers is a good review subject. In this regard, the present review manuscript will be good for the Nanomaterials.

Response 1: We are of great gratitude to your positive appraisal and instructive comments. We have substantially revised the manuscript according to your following comments.

Point 2: The scientific principles and strategies to improve the photocatalytic performance in this manuscript are conventional and those are common in inorganic photocatalytic materials. In my opinion, the metal doping, semiconductor heterojunction, and defect effects are very common in inorganic photocatalysts. Considering the intention of authors, are there special benefits of the electrospun inorganic photocatalysts in the aspect of principles or in the aspect of synthetic methods? If the authors can supply the special benefits of the electrospun inorganic photocatalysts, compared with the conventional inorganic photocatalysts, those will be very helpful for the readers.

Response 2: Thank you for the kind suggestion. We have supplied the benefits of the electrospun inorganic photocatalysts compared with the conventional inorganic photocatalysts in the revised manuscript.

Point 3: If the authors can add the summary table for the reported electrospun inorganic photocatalysts, features, strategies, and the comparative performance, those will be very helpful to follow the contents of this manuscript.

Response 3: Thank you for your instructive suggestion. We have provided the summary table (Table 1 in the revised manuscript) for the reported electrospun inorganic photocatalysts, features, strategies, and the comparative performance.

Reviewer 3 Report

The proposed manuscript is devoted to the adaptation of photocatalysts to solve global ecological and energy problems. This topic is of high interest and high importance and now-a-days many study results are presented in this field. Thus, to propose a review paper devoted this topic, is quite reasonable. The manuscript contains lot of information, and can be helpful for the researchers working or starting to work in this field.

Nevertheless, before to recommend the manuscript for publication I would like to see some additions and modifications. The point is that the authors focus the listing of different results obtained by various researchers in the last time, whereas the explanations of the peculiarities of the processes providing these results, are restricted only by short remarks, if any. Thus, the review is similar to a short handbook. In my opinion, in addition to helpful information, the review paper should present some ideas determining and providing the progress in the field in question, as well as demonstrate the advantages of the developed methods, discussed in the manuscript.

Besides, it is suitable to demonstrate why the ceramic fibers, fabricated by electrospinning with subsequent post-processing, are the best way for declared goals.

In addition, regarding the electrospinning, I have to remark that, unfortunately, the authors understanding of the process mechanisms is not so proper.

Indeed, from the beginning when the authors try to explain the electrospinning basics, I see incorrect statements or inaccurate sentences:

1. Page 1, line 25

"Electrospinning is based on inducing static electrical charges on the molecules of a solution and then the self-repulsion of the charges causes the liquid to stretch into a fiber in an electric field [1-3]."

The physical mechanism providing the electrospinning process is absolutely else, being related to the interaction of polarized polymer solution with external electrostatic field.

2. Page 1, line 25

"Since Formalas issued the first electrospinning patent in 1934 [4], this technology did not receive much attention until the research boom of nanomaterials in the 1990s."

In reality, the first electrospinning patent was by J. F. Cooley in 1902: US 692631 4th February, 1902 "Apparatus for electrically dispersing fluids."

3. Page 3, line 111

"At proper voltage, the self-repulsion of the charges causes the liquid to stretch into a fiber…"

See 1item.

In some cases, the authors comments are too short, and some information presented in the cited plots, remains unclear. For example,

Page 4. Figure 3.

What is the meaning of the curve P25?

Page 7. Figure 5c.

Only curve C2 is mentioned in the text, and I found nothing regarding other curves.

I hope that after a necessary revision, the proposed manuscript can be recommended for publication.

Round 2

Reviewer 1 Report

The paper can be accepted.